# Transcriptomics and Proteomics Characterizing the Anticancer Mechanisms of Natural Rebeccamycin Analog Loonamycin in Breast Cancer Cells

**DOI:** 10.3390/molecules27206958

**Published:** 2022-10-17

**Authors:** Xiao Sun, Zhanying Lu, Zhenzhen Liang, Bowen Deng, Yuping Zhu, Jing Shi, Xiaoling Lu

**Affiliations:** 1Experimental Training Center of Basic Medical Science, College of Basic Medical Sciences, Naval Medical University, Shanghai 200433, China; 2Department of Biochemistry and Molecular Biology, College of Basic Medical Sciences, Naval Medical University, Shanghai 200433, China; 3State Key Laboratory of Pharmaceutical Biotechnology, Chemistry and Biomedicine Innovation Center (ChemBIC), School of Life Sciences, Nanjing University, Nanjing 210023, China

**Keywords:** loonamycin, triple negative breast cancer, p53, PI3K/AKT/mTOR, O-glycan

## Abstract

The present study is to explore the anticancer effect of loonamycin (LM) in vitro and in vivo, and investigate the underlying mechanism with combined multi-omics. LM exhibited anticancer activity in human triple negative breast cancer cells by promoting cell apoptosis. LM administration inhibited the growth of MDA-MB-468 tumors in a murine xenograft model of breast cancer. Mechanistic studies suggested that LM could inhibit the topoisomerase I in a dose-dependent manner in vitro experiments. Combined with the transcriptomics and proteomic analysis, LM has a significant effect on O-glycan, p53-related signal pathway and EGFR/PI3K/AKT/mTOR signal pathway in enrichment of the KEGG pathway. The GSEA data also suggests that the TNBC cells treated with LM may be regulated by p53, O-glycan and EGFR/PI3K/AKT/mTOR signaling pathway. Taken together, our findings predicted that LM may target p53 and EGFR/PI3K/AKT/mTOR signaling pathway, inhibiting topoisomerase to exhibit its anticancer effect.

## 1. Introduction

Triple negative breast cancer (TNBC) is a special subtype of breast cancer, accounting for about 12.7% of breast cancer [1], which is characterized by negative estrogen receptor (ER), progesterone receptor (PR) and human epidermal growth factor receptor-2 (HER2) with high recurrence rate, strong invasiveness, and poor prognosis characteristics. Due to the lack of corresponding targets and high heterogeneity, the treatment of TNBC is mainly limited to chemotherapy [2]. It is valuable to develop novel chemotherapeutics that can broadly target TNBCs to render this highly deadly disease subtype curable.

Marine natural products provide an important source of lead compounds for new drug research and development because of their unique structure and diversity biological activity. At present, more than 35,000 marine natural compounds have been discovered in the world, most of which possess unique structures and present diverse biological activity. Loonamycin (LM) is an indole carbazole compound rebeccamycin analog produced from *Nocardiopsis flavescens* NA01583 isolated from marine sediment in Yongxing Island, South China Sea [3] (Figure 1A). To the best of our knowledge, rebeccamycin is a cytotoxicity compound that binds to topoisomerase I to inhibit the reconnection at the DNA strand incision [4], leading to the break of DNA single strand and double strand. This compound showed an impressive cytotoxicity in vitro but could not be further developed because of poor water solubility. Some rebeccamycin analogues (e.g., becatecarin [5] and edotecacin [6]) have entered clinical research. LM has a rare sugar group and hydroxyl group compared with rebecamycin [3], which could increase its water solubility, and improve its pharmaceutical potential.

Our previous studies showed that LM had strong cytotoxic activities against various tumor cell lines, especially to the human TNBC cell line MDA-MB-468. In this article, we used RNA-seq and TMT quantitative proteomics technology to uncover the mechanism of LM in TNBC cells.

## 2. Results

### 2.1. LM Targets TNBC In Vitro and In Vivo

First, LM was investigated for its inhibitory effects on several breast cancer cell lines (Table 1). LM displayed a preferential anticancer activity against the cell lines with IC_50_ values of 0.517 μM, 0.197 μM and 0.372 μM in the MCF-7 cell line, MDA-MB-231 cell line and MDA-MB-468 cell line. The IC_50_ of LM for LO2 was 1.022 μM. The LO2 was used as a control cell line and the result suggested that LM was less toxic to LO2 than the cancer cell line. Then, we further investigated the inhibitory effects of LM on TNBC MDA-MB-468 cells. As exhibited in Figure 1B, LM displayed a good anticancer activity against MDA-MB-468 in a dose-dependent manner with IC_50_ values of 0.372 μM and 0.235 μM after 48 h and 72 h.

Second, we examined the effect of LM on cell cycle in MDA-MB-468 cells. In MDA-MB-468 cells, after 48 h of incubation, LM induced cell cycle arrest in the G2 phase in a dose-dependent manner (Figure 1C). At 1.6 μM LM, the value of G1-phase declined to 24.46%, while G2-phase increased to 48.90%. The values of the S phase showed a slight change. Growing lines of evidence indicate that eukaryotic topoisomerase activity is monitored and regulated throughout the cell cycle [7]. As rebeccamycin is a cytotoxicity compound which is bound to the topoisomerase I, we detected the inhibitory effect of LM on topoisomerase I activity (Figure 1D,E). Topoisomerase I (Topo I) relaxes the super helix structure of DNA by cutting its single strands [8]. The normal plasmid DNA is a closed double stranded DNA. In the process of electrophoresis, the plasmid may have three configurations: the superhelical DNA (SC DNA), the open circular DNA (OC DNA) and the linear DNA (L DNA). The SC DNA is at the front of gel, OC DNA is at the back, and L DNA is between SC DNA and OC DNA. After the cleavage of topoisomerase I, the structure of SC DNA will be destroyed. The experiment showed that LM could inhibit topoisomerase I activity in a dose-dependent manner.

Next, the effects of LM on the growth and formation of subcutaneous xenograft nodes derived from the inoculated MDA-MB-468 cells in vivo in BALB/c nude mice were investigated. Both the volumes and weights of the formed MDA-MB-468 cells tumor nodes were reduced by LM administration every 2 days for a total of 17 days at a concentration of 10 mg/kg or 20 mg/kg LM by i.v., compared to 15 mg/kg for paclitaxel as a positive control [9] by i.p. and 2% DMSO as vehicle by i.v. The low-dose group was of no statistical significance compared with the negative control group. The volumes and weights of tumors in high-dose group reduced significantly (Figure 1F–I). The frozen section and HE staining on the stripped tumor tissue was performed, and obvious tumor-like tissues were observed under the microscope, such as large cell volume, big nucleus and deformity loose arrangement of tumor cells (Figure 1J). There were no detectable toxic or necrotic effects on the heart, liver, spleen, lung or kidney tissues after LM treatment and no significant weight loss. Taken together, these data suggest that LM exhibits good therapeutic activity.

### 2.2. Functional Annotation Enrichment of LM-Regulated Genes

To uncover the LM regulatory mechanism in TNBC MDA-MB-468 cells, we performed RNA-seq analysis to profile the transcriptomes of MDA-MB-468 cells when treated with 1.6 μM LM. Differentially abundant genes (DAGs) were those meeting the qualified data (fold change ≥ 1.2 and *p* < 0.05) under comparison of LM vs. the control group. A total of 1764 DAGs were shown in the volcano map, of which 737 genes were upregulated and 1027 genes were down-regulated (Figure 2A). GO (Gene ontology) is a comprehensive database that describes gene functions. GO enrichment analysis basing on the DAGs including up-regulated genes and down-regulated genes, were mapped the differential genes to the entries in the three aspects of cell components (CC), molecular functions (MF), and biological processes (BP) (Figure 2B). Through GO enrichment analysis, we can roughly understand which biological functions, signal pathways, or cell locations are enriched of the DAGs. In terms of the CC, the up-regulated genes were mainly enriched in the extracellular matrix, anchored component of membrane, collagen trimer and MHC protein complex. The down-regulated genes were mainly enriched in postsynapse, synaptic membrane, receptor complex, ion channel complex. In terms of the MF, the up-regulated genes were mainly enriched in the receptor ligand activity, receptor regulator activity, cytokine receptor binding, extracellular matrix structural constituent. The down-regulated genes were mainly enriched in actin binding, ion gated channel activity, Ras GTPase binding, passive transmembrane transporter activity. In terms of the BP, the up-regulated genes were mainly enriched in the regulation of signaling receptor activity, regulation of endothelial cell proliferation, endothelial cell proliferation, positive regulation of locomotion and wound healing. The down-regulated genes were mainly enriched in trans-synaptic signaling, chemical synaptic transmission, anterograde trans-synaptic signaling, regulation of membrane potential and neurotransmitter levels.

KEGG (Kyoto Encyclopedia of Genes and Genomes) is a comprehensive database integrating genome, chemistry and system function information. It stores information on gene pathways of different species. KEGG pathway enrichment analysis was conducted to describe the significant changes in signal pathways of DAGs (Figure 2C). The results showed that the up-regulated differential genes were mainly centered on the pathways in cancer, hippo signaling pathway, p53 signaling pathway, etc. The down-regulated proteins were mainly concentrated on the glucagon signaling pathway, calcium signaling pathway, phosphatidylinositol signaling system, propanoate metabolism, inositol phosphate metabolism, pyruvate metabolism, other types of O-glycan biosynthesis and so on.

GSEA enrichment analysis was conducted to explore the changes of gene expression in the pathway and find the upstream factors leading to these changes(Figure 2D). In our studies, the curated gene sets as the exploration set showed that p53 and downstream signal pathway were up-regulated, and O-glycan and PI3K-AKT-mTOR signaling pathway were down-regulated in LM treatment group.

### 2.3. Proteomic Expression Profiling of LM-Treated TNBC Cells

To further elucidate cellular mechanism and molecular function, we performed TMT quantitative proteomics analysis to assess the protein expression profiles in MDA-MB-468 cells treated with 1.6 μM LM. Differentially abundant proteins (DAPs) were those meeting the qualified data (fold change ≥ 1.2 and *p* < 0.05) under comparison of LM vs. control group. A total of 1314 DAPs were shown in the volcano map, of which 778 proteins were upregulated and 536 proteins were down-regulated (Figure 3A). Wolf PSORT software was used for localization analysis of differential proteins, which show that the DAPs were mainly distributed in cytoplasm, nucleus, mitochondria, and plasma membrane (Figure 3B). In total, 119 GO terms were obtained based on the DAPs including up-regulated proteins and down-regulated proteins (Figure 3C). GO enrichment analysis showed that in terms of the cell components (CC), the up-regulated proteins were mainly enriched in the cytosol (GO:0005829), ficolin-1-rich granule lumen (GO:1904813) and nucleus (GO:0005634). The down-regulated proteins were mainly enriched in the mitochondrial matrix (GO:0005759), integral component of plasma membrane, (GO:0005887) and nBAF complex (GO:0071565). In terms of biological process (BP), the up-regulated proteins were mainly enriched in the processes related to gluconeogenesis (GO:0006094), negative regulation of ryanodine-sensitive calcium-release channel activity (GO:0060315) and proteasomal ubiquitin-independent protein catabolic process (GO:0010499). The down-regulated proteins were mainly enriched in the processes related to isoleucine catabolic process (GO:0006550), O-glycan processing (GO:0016266) and leucine catabolic process (GO:0006552). In terms of molecular function (MF), the up-regulated proteins were mainly enriched in S100 protein binding (GO:0044548), RAGE receptor binding (GO:0050786) and threonine-type endopeptidase activity (GO:0004298). The down-regulated proteins were mainly enriched in polypeptide N-acetylgalactosaminyl transferase activity (GO:0004653), biotin binding (GO:0009374) and signaling receptor activity (GO:0038023).

KEGG pathway enrichment analysis was conducted to describe the significant changes in pathways of DAPs (Figure 3D). The results showed that the DAPs including up-regulated proteins and down-regulated proteins were classified into 34 terms. The up-regulated differential proteins were mainly centered on the changes of metabolic pathway, like amino acid biosynthesis, glucose metabolism, nucleotide metabolism, glutathione metabolism, and p53 pathway, etc. The down-regulated proteins were mainly concentrated on the pathway of amino acid degradation, mTOR signaling pathway, PI3K-Akt signaling pathway, O-glycan biosynthesis, etc. Among these DAPs, the expressions of classical tumor related signaling pathway p53 were up-regulated and the EGFR/mTOR pathway were down-regulated.

GSEA enrichment analysis was conducted to explore the changes of gene expression in the pathway and find the upstream factors leading to these changes (Figure 3E). In our studies, the curated gene sets as the exploration set showed that p53 and downstream signal pathway were up-regulated, and the EGFR and mTOR related pathway, and O-glycan were down-regulated in LM treatment group.

### 2.4. Validation of Transcriptomic and Proteomic Results

According to the combined analysis of RNA-seq and TMT-based quantitative proteomic, it is suggested that LM may have a significant effect on O-glycan, p53-related signal pathway and EGFR/PI3K/AKT/mTOR signal pathway in enrichment of the KEGG pathway. The GSEA data also suggests that the TNBC cells treated with LM may be regulated by p53 and EGFR/PI3K/AKT/mTOR signaling pathway.

The Cys124 located in the loop1/sheet3 (L1/S3) pocket of the p53 protein plays a key role in maintaining p53 stable conformation (Figure 4A). The covalently binding model between LM and p53 (L1/S3) was investigated by molecular docking. The binding energy was −26.8 kJ/mol. The main combination modes were the hydrogen bonds, as shown in Figure 4D. The ether bond of LM formed hydrogen bond with Thr102 of p53. The methoxy group on the rare sugar group of LM formed hydrogen bond with Phe113 of p53. The hydroxyl group of LM formed one hydrogen bond with Leu114, His115, Cys124 and His 233 separately. The hydroxyl and ether group of LM formed three hydrogen bonds with Thr123. The above results suggested that LM can target the pockets of wild-type p53 l1/s3, improving the stability of p53 and activating p53 related pathways.

To verify the combined analysis of transcriptomic and proteomic results, the western blot of key proteins was performed. It is determined that expression of p53 was increased after treated with LM for 48 h on MDA-MB-468 (Figure 4B,C), but there was no statistical significance. The expressions of EGFR, PI3K, mTOR and BCL-2 were significantly down-regulated (Figure 4B,D). The expression of p-p53 was significantly up-regulated (Figure 4B,C), and the expression of p-EGFR, p-PI3K and p-mTOR were significantly down-regulated (Figure 4B,D). It was shown that LM could activate p53 and inhibit the expression of EGFR, PI3K, mTOR and BCL-2.

## 3. Discussion

In this study, we evaluated the anticancer activity of LM, an indole carbazole rebeccamycin analog from *Nocardiopsis flavescens* NA01583. Our data showed that LM could inhibit tumor cell growth both in vitro and in vivo. The inhibitory effect of LM in cancer cells demonstrated that it could inhibit the topoisomerase I in a dose-dependent manner in vitro experiments like other rebeccamycin analogs [10]. Topo I cut one strand of DNA to form single-strand breaks, allowing supercoiled DNA to relax, otherwise it can hinder DNA replication and transcription, and thus block cell growth [11,12,13]. For rapid cell division, cancer cells need high Topo I activity to finish the DNA metabolic processes. Topoisomerase I inhibitors can block the reconnection of DNA strands, lead to the accumulation of Topo I-breaking complexes, inhibit replication and transcription, and cause DNA damage, thus activating DNA damage checkpoints and inhibiting the progress of cell cycle [14]. Topoisomerase were recognized as promising targets in cancer, and various DNA Topos inhibitors have been on the market [15]. Some rebeccamycin analogues like becatecarin and edotecacin have entered clinical research. Becatecarin intercalates into DNA and inhibits the catalytic activity of topoisomerases I/II [16]. Edotecarin is a potent inhibitor of topoisomerase I and also has an effect on protein kinase C [17].

Next, the RNA-Seq and TMT were conducted to investigate the changes in transcriptomic and proteomic profile of MDA-MB-468 cells treated with LM (LM). The combined transcriptome and proteome analyses revealed that LM may activate p53 and inhibit the O-glycans, inhibiting EGFR/PI3K/mTOR signaling pathway to exhibit its cytotoxicity activity. The tumor suppressor p53 functions mainly as a transcription factor. A mutation of the TP53 gene that encodes p53 protein is the main way of inactivating p53. p53 is a key tumor suppressor in the process of preventing tumorigenesis. The dysfunction of p53 often leads to cancer. When cells suffer from DNA damage, excessive proliferation, hypoxia, lack of nutrition, telomere loss, or in the environment of oxidative stress, lack of nucleotides or replication pressure, p53 will be activated to induce cell cycle arrest, apoptosis, aging or autophagy, preventing cells from growing and dividing and killing cells before they become cancerous [18]. p53 removes cells with high mutation risk in this way, inhibiting tumor formation. In normal cells, p53 remains at a low level and dormant state under non-stress conditions to prevent its adverse effects on cell growth. Its low expression is mainly by its interaction with ubiquitin E3 ligase MDM2 [19]. A stress response can prevent MDM2 mediated p53 degradation by various mechanisms, promoting the stability and activation of p53. About 50% of cancers still express wild-type p53, but these p53 proteins usually lose its function, because of the over activation of MDM2 and MDMX. The stability and transcriptional activity of p53 depend on its phosphorylation [20]. According to research, the phosphorylation of p53 protein Ser15, Ser20, Ser33 and Ser37 could enhance its binding to P300, thereby activating the transcriptional activity of p53.The phosphorylation of Thr18 can not only enhance p53-p300 binding, but also interfere with p53-MDM2 binding [21]. Reactivating p53 and restoring its function is a feasible and promising tumor treatment strategy. Moreover, p53 is the most common mutant gene in human cancer, of which p53 mutations are found in more than 50% of tumors. For example, approximately 80% of TNBCs express an inactive, mutant form of the p53 tumor suppressor protein (mtp53), resulting in rapid tumor growth and metastasis [22]. Many mutations occur in the DNA binding domain of p53 gene and the altered mutant p53 protein (mtp53) is subsequently not degraded, in which high levels of mtp53 protein accumulate within the cell, leading to the development of tumors. Therefore, converting the mtp53 protein back into its functional wild-type conformation is also a promising means to prevent or reverse tumor development. Restoring the function of wild-type p53 and developing drug candidates for mutant p53 to restore the normal function of p53 can activate p53 to inhibit tumor. At present, many anti-cancer drugs targeting p53 have been developed. For example, the arsenic trioxide, which is used for acute primary myeloid leukemia, can restore the transcriptional activity of p53 mutants by arsenic ions binding to three cysteine residues in the DNA-binding domain [23]. HDAC6 (Histone deacetylase 6) can promote the combination of HSP90 (heat shock protein 90) and mutant p53 protein by catalyzing HSP90 deacetylation and thus make mutant p53 molecule more stable. So, some HDAC6 inhibitors (e.g., statin) or HSP90 inhibitors (e.g., Ganetespib) have been found to induce mutant p53 degradation [24,25]. Small molecule compounds like RG7112 and some nucleic acid drugs can activate p53 by promoting the expression of TP53 gene or inhibiting the expression of MDM2 [26,27,28].

In 2002, PRIMA-1 was discovered as a mutant p53 reactivator based on the tumor cells screening of the mutated p53 [29]. The specific mechanism was not studied that it directly bound to the Cys124 of mutant p53 protein until 2009 [30]. Cys124 locating in loop1/sheet3 (L1/S3) pocket of p53 protein, plays a key role in maintaining stable conformation. Loop1 can directly interact with helix2 in the p53 DNA binding domain, suggesting that the stability of p53 can be improved by small molecules compounds to stabilize loop1. The compounds targeting wild-type p53 L1/S3 pockets could improve the stability of p53. L1/S3 pocket was a target for pharmaceutical reactivation of p53 mutants [31]. We constructed the computer virtual screening system with wild-type p53 as the target. In the computer docking experiment, LM could bind with L1/S3 pockets of p53 protein well, forming one hydrogen bond with Cys124. Subsequent WB experiments showed that LM could activate p53 signal pathway. Although, the molecular docking did not provide direct proof that LM interacts with p53, but explored a potential route to influence the p53 pathways.

The epidermal growth factor receptor (EGFR) is a receptor tyrosine kinase that belongs to the ErbB family and is involved in angiogenesis, cell proliferation, metastases as well as inhibition of apoptosis. It was demonstrated that EGFR is overexpressed in TNBC cells [32]. Its expression was an independent poor prognostic factor associated with worse DFS and OS [33,34]. In light of the high expression level of EGFR and its strong effect on cell proliferation and motility, EGFR has been considered as an attractive therapeutic target for TNBC [35]. EGFR is on the upstream of PI3K and activates PI3K/AKT/mTOR pathway. The PI3K/AKT/mTOR pathway is associated with cell metabolism, proliferation, differentiation, and survival. PI3Ks are heterodimers composed of regulatory (p85) and catalytic (p110) subunits and exist in four isoforms (α, β, δ, and γ) [36]. The signaling pathway is activated by stimulation of receptor tyrosine kinases, which in turn trigger PI3K activation, followed by phosphorylation of AKT and mTOR complex 1 (mTORC1). It is speculated that on one hand, LM could restore the function of wild-type p53 to activate p53, on the other hand, LM may inhibit the classical EGFR/PI3K/AKT/mTOR pathway to inhibit the growth of the cells; the specific mechanism needs to be further explored.

Glycosylation is one of the most important post-translational modifications of the protein, including N-glycosylation and O-glycosylation. O-linked glycosylation is considered more complicated than N-linked for its unknown initiation [37]. O-glycosylation added single monosaccharides one by one through enzymatic reaction. The linking monosaccharide GalNAc is added directly to Ser/Thr/Tyr residues in glycoproteins within the Golgi apparatus from the nucleotide sugar donor uridine diphospho-GalNAc (UDP-GalNAc). This linking sugar is commonly modified in all cells by the addition of galactose (Gal) from the donor UDP-Gal to create the disaccharide Galβ1-3GalNAcα1-O-Ser/Thr, known as core 1. Such O-glycans can be further modified and extended within the Golgi apparatus to generate an incredible diversity of many tens of thousands of different glycan structures. Meanwhile, the changes in the core structure of several types of O-glycans are related to multiple cancers, which abnormal O-linked glycosylation has been widely proved to act biological functions to directly result in cancer growth and progression. The T antigen and sialyl-Tn antigen (STn), tumor-associated carbohydrate antigens (TACAs), are truncated O-glycans commonly expressed by carcinomas on multiple glycoproteins which serve as potential biomarkers for tumor presence and stage both in immunohistochemistry and in serum diagnostics [38]. CA199 and CA125 are used as circulating tumor biomarkers. In 90% of breast cancers, altered O-glycosylation has been observed to have a correlation with cancer progression, worse prognosis, and metastatic potential; like the number of O-GalNAc glycans in glycoproteins changes, the core structure of O-GalNAc glycosylation changes, and breast cancer cells with shorter O-glycans (abnormal glycosyltransferase activity, premature sialylation of polylactosamine chain blocking the addition of more glycans or truncation of O-glycans at core 1 level). These abnormalities lead to the expression of TACA, such as Tn antigen, St antigen and STn antigen [39]. The polypeptide-N-acetylgalactosaminyl transferase (GT) is a key enzyme of O-linked glycosylation. In our validation experiment, LM down-regulated the GalNAc-T2 with no statistical difference. It is speculated that some other members of GTs should be validated in the future.

In general, the study showed that LM was a potential antitumor compound in vitro and in vivo. LM may target p53 and EGFR/PI3K/AKT/mTOR signaling pathway, inhibiting topoisomerase to exhibit its anticancer activity according to the combined transcriptome and proteome analyses. Our study provided important information for the specific cytotoxicity mechanism of LM and explanation of the anticancer activity of rebeccamycin analog.

## 4. Materials and Methods

### 4.1. Reagents

LM was isolated from Nocardiopsis flavescens NA01583 from marine sediment and provided by research group of Prof. Ge Huiming, Nanjing University. It was dissolved in dimethyl sulphoxide and stored at −20 °C until use. The Cell Cycle Analysis Kit, penicillin/streptomycin were acquired from Beyotime. The fetal bovine serum (FBS), phosphate-buffered saline (PBS), Dulbecco’s modified Eagle medium (DMEM) and Leibovitz’s L-15 medium (L-15) were obtained from Gibco (Thermo Fisher Scientific, Waltham, MA, USA). The cell counting kit-8 (CCK8) was purchased from Dojindo Molecular Technology. Specific primary antibodies against β-actin, mTOR, P-mTOR, PI3K, P-PI3K, EGFR, p-EGFR, AKT, P-AKT, BCL-2, p53, p-p53 were acquired from CST (Cell Signaling Technology, Danvers, MA, USA); 0.25% trypsin and 0.2% EDTA were purchased from Gibco (Thermo Fisher Scientific, Waltham, MA, USA).

### 4.2. Cell Culture

TNBC cell line MDA-MB-468 (Cat.TCHu136) were cultured in 90% L-15 medium supplemented with 10% fetal bovine serum and 100 mg/L streptomycin–100 U/mL penicillin mixture. Cells were cultured at 37 °C in an incubator with controlled humidified atmosphere. The cell dispersed liquid was prepared by 0.25% trypsin plus 0.2% EDTA for subculturing and then spun down by centrifugation at 800 rpm for 5 min, after which the supernatant was removed and precipitated cells was resuspended in culture medium.

### 4.3. Growth Curve Measured by CCK-8 Method

The MDA-MB-468 cells were seeded in 96-well E-plates with the density of 5000 cells/well. After 24 h, the cells were treated with LM or 0.06% DMSO as control for 48 h or 72 h. LM treatment concentrations were 0.0625 μM, 0.125 μM, 0.25 μM, 0.5 μM, 1 μM, 2 μM and 4 μM, and each group was performed in triplicate; 10 μL CCK-8 reagent was added for 1 h after treatment and the optical density value (OD_450_) was measured at the wavelength of 450 nm. The IC50 of LM was calculated by the cell growth curves drawn with Prism-GraphPad.

### 4.4. Topoisomerase I-Mediated DNA Relaxation and Cleavage Assays

The different concentrations of LM (5 μM, 10 μM, 20 μM) and camptothecin (CPT, 20 M) were incubated with supercoiled pUC19 plasmid DNA in relaxation buffer (50 mM Tris-HCl (pH7.5), 100 mM KCl, 0.5 mM EDTA and 30 pg/mL BSA) for 15 min at 37 °C to ensure binding equilibrium. Then, the recombinant topoisomerase I enzyme (from calf thymus, Beytime, China) was added for a further 30 min of incubation at 37 °C. The mixture of sodium dodecyl sulfate (SDS) and protease K (the final concentration was 0.25% and 250 g/mL, respectively) was added for a 30 min incubation at 50 °C to terminate the reaction. To obtain single stranded DNA, samples were loaded onto a 1% agarose gel lacking ethidium bromide at room temperature for 2 h at 120 V in TBE buffer. Gels were stained after migration using Gelred and then washed and finally photographed under UV light.

### 4.5. RNA-seq Analysis

Total RNA was extracted from cells using TRIzol^®^ reagent, and genomic DNA was removed using DNase I. RNA integrity was detected by Agilent 2100 BioAnalyzer. The library was built by the NEB method. AMPure XP Beads were used to screen cDNA, conduct PCR amplification, and purify PCR products. NEBNext^®^ Ultra™ RNA Library Prep Kit for Illumina^®^ was used for Library construction. The library was initially quantified by Qubit2.0 Fluorometer and was diluted to 1.5 ng/uL. The insert size of the library was detected using Agilent 2100 BioAnalyzer. qRT-PCR was used to quantify the effective concentration of the library to ensure the quality of the library. Then, the Illumina sequencing was conducted. The basic principle is sequencing by synthesis.

The raw data was filtered by the removal of reads with adapter, the removal of reads with N (N indicates that the base information cannot be determined), and the removal of low-quality reads. At the same time, the Q20, Q30 and GC contents of clean data were calculated. All subsequent analyses were based on clean data. Hisat2v2.0.5 was used to compare clean reads of paired terminal with genomic species: human genes (GrCH38.p12). String Tie (1.3.3b) (Mihaela-Pertea et al., 2015) was used for new gene prediction. Feature Counts (1.5.0-P3) were used to calculate recounts mapped to each gene. The FPKM of each gene was calculated based on the length of the gene and the readout mapped to the gene was calculated to obtain the expression value matrix.

Gene expression analysis of the different groups was performed by The DESeq2 software (V1.16.1) (*n* = 3). The *p*-value was adjusted using Benjamini and Hochberg’s method. Genes with an adjusted *p*-value (FDR) < 0.05 were defined as differentially expressed genes (DEGs). Gene Ontology analysis (GO) and KEGG pathway enrichment analysis of DEGs were implemented by cluster Profiler (3.4.4). GSEA enrichment analysis was performed using GSEA (V4.1.0).

### 4.6. TMT Labeling and LC-MS/MS Analysis

The MDA-MB-468 cells were treated with 1.6 μM LM or 0.06% DMSO control medium for 48 h. SDT buffer was added to the sample. The lysate was sonicated and then boiled for 15 min. After centrifuged at 14,000× *g* for 40 min, the supernatant was quantified with the BCA Protein Assay Kit (P0012, Beyotime, Shanghai, China). The protein was digested by Filter aided proteome preparation (FASP) method, and 100 μg peptide mixture of each sample was labeled using TMT reagent according to the manufacturer’s instructions (Thermo Fisher Scientific, USA). TMT labeled peptides were fractionated by RP chromatography using the Agilent 1260 infinity II HPLC. The collected fractions were combined into 10 fractions and dried down via vacuum centrifugation at 45 °C.

Each fraction was injected for nano LC-MS/MS analysis. The peptide mixture was loaded onto the C18-reversed phase analytical column (Thermo Fisher Scientific, Acclaim PepMap RSLC 50 μm × 15 cm, nano viper, P/N164943) in buffer A (0.1% Formic acid) and separated with a linear gradient of buffer B (80% acetonitrile and 0.1% Formic acid) at a flow rate of 300 nL/min. The linear gradient was as follows: 6% buffer B for 3 min, 6–28% buffer B for 42 min, 28–38% buffer B for 5 min, 38–100% buffer B for 5 min, hold in 100% buffer B for 5 min.

LC-MS/MS analysis was performed on a Q Exactive HF mass spectrometer (Thermo Fisher Scientific, Waltham, MA, USA) that was coupled to Easy nLC (Thermo Fisher Scientific, Waltham, MA, USA) for 60 min. The mass spectrometer was operated in positive ion mode. MS data was acquired using a data-dependent top 10 method dynamically choosing the most abundant precursor ions from the survey scan (350–1800 *m*/*z*) for HCD fragmentation. Survey scans were acquired at a resolution of 60,000 at *m*/*z* 200 with an AGC target of 3 × 10^6^ and a maxIT of 50 ms. MS2 scans were acquired at a resolution of 15,000 for HCD spectra at *m*/*z* 200 with an AGC target of 2 × 10^5^ and a maxIT of 45 ms, and the isolation width was 2 *m*/*z*. Only ions with a charge state between 2 and 6 and a minimum intensity of 2 × 10^3^ were selected for fragmentation. Dynamic exclusion for selected ions was 30 s. Normalized collision energy was 30 eV.

MS/MS raw files were processed using MASCOT engine (Matrix Science, London, UK; version 2.6) embedded into Proteome Discoverer 2.2. The protein database was Uniprot_HomoSapiens_20367_20200226. The search parameters included trypsin as the enzyme used to generate peptides with a maximum of 2 missed cleavages permitted. A precursor mass tolerance of 10 ppm was specified and 0.05 Da tolerance for MS2 fragments. Except for TMT labels, carbamidomethyl (C) was set as a fixed modification. Variable modifications were Oxidation(M) and Acetyl (Protein N-term). A peptide and protein false discovery rate of 1% was enforced using a reverse database search strategy. Proteins with fold change > 1.2 and *p* value (Student’s *t*-test) < 0.05 were considered to be differentially expressed proteins. Wolf PSORT software was used for localization analysis of differential proteins. Gene Ontology analysis (GO) and KEGG pathway enrichment analysis of DEGs were implemented by clusterProfiler (3.4.4). GSEA enrichment analysis was performed using GSEA (V4.1.0).

### 4.7. Molecular Docking

Molecular docking was performed using the Schrodinger software (Schrödinger, Inc., New York, NY, USA). The 3D structure of p53 were retrieved from the protein data bank (PDB ID: 1TSR). In LM-p53 covalent docking, the protein p53 was prepared by Protein Preparation Tool in Schrodinger including optimized hydrogen bond network at pH 7.0 with PROKA tool. The ligand LM was prepared by Avogadro Tool to obtain the structural optimization.

### 4.8. Western Blot Analysis

The MDA-MB-468 cells were treated with 1.6 μM LM or 0.06% DMSO control medium for 48 h. Cells were harvested with trypsin/EDTA and then total proteins were extracted by using RIPA lysis buffer (RIPA lysis buffer (Beyotime, Shanghai, China)), 100 μg/mL PMSF (Beyotime, Shanghai, China), and 1 × protease inhibitor (Sigma, St. Louis, MO, USA). The protein was quantified by BCA methods according to the instructions (Beyotime, Shanghai, China). The proteins extracted from cells were separated by SDS-PAGE electrophoresis and transferred to nitrocellulose membrane. The membrane was incubated with primary antibodies and anti-rabbit IgG (HRP-linked). The bands were detected by ECL Western blot system (Kodak, Rochester, NY, USA). Western blotting bands from three independent measurements were quantified with ImageJ.

### 4.9. Xenograft Model

Twenty-five female BALB/c nude mice aged 6–8 weeks were kept at constant temperature and humidity. The body weight was 20–24 g. Animals were supplied by Laboratory Animal Business Department of Shanghai Family Planning (certificate of quality: 20210715Abzz0619000729). Each mouse was inoculated subcutaneously at the right flank with MDA-MB-468 tumor cells (1 × 10^7^) in 0.2 mL of PBS with Matrigel (1:1) for tumor development. Treatments were started on day 27 after tumor inoculation when the average tumor size reached 160 mm^3^. The animals were assigned into groups randomly based upon tumor volumes. Each group consisted of 4 tumor-bearing mice. The vehicle was 2%DMSO + 15% Solutol + 83%Saline. The mice in control group were injected 2.5 mL/kg vehicle by i.v. The mice in the positive control group were injected 15 mL/kg Paclitaxel by i.p. The mice in the low-dose group were injected 10 mL/kg LM by i.v. The mice in the high dose group were injected 20 mL/kg LM by i.v. The animals were checked daily for any effects of tumor growth and treatments on normal behavior, and body weights were measured every 3 days. Tumor size was measured every 3 days by a caliper using the formula: V = 0.5a × b^2^ where a and b are the long and short diameters of the tumor, respectively.

### 4.10. Statistical Analysis

One-way ANOVA analysis was used for comparison between groups, and Student’s t-test was used for pair-wise comparison within groups. All statistical analyses were processed with GraphPad Prism 7.0 software (GraphPad Software, San Diego, CA, USA). *p* < 0.05 was considered statistically significant.

## Figures and Tables

**Figure 1 molecules-27-06958-f001:**
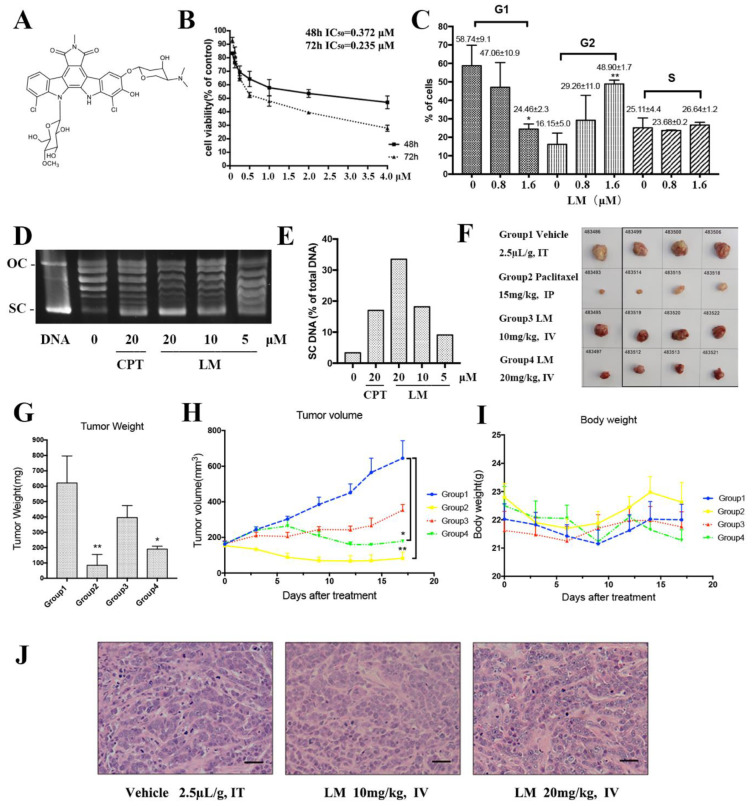
LM targets TNBC in vitro and in vivo. (**A**) The structure of LM; (**B**) the cell viability experiment of LM on human TNBC cell line MDA-MB-468. The MDA-MB-468 cells were treated with 0, 0.0625, 0.125, 0.25, 0.5, 1, 2,4 µM LM for 48 and 72 h, and the cell viability was detected by the CCK8 assay; (**C**) effects of LM at 0.8 μM and 1.6 μM on MDA-MB-468 cell cycle arrest after 48 h of incubation. Cells were stained with Annexin V and analyzed by flow cytometry. The vertical bars represent the standard deviation of means (SD) (*n* = 3 experiments); * *p* < 0.05 and ** *p* < 0.01, vs. negative control. The G1, G2 and S represent the phase of the cell cycle. (**D**) The inhibitory effect of LM on topoisomerase I activity. The plasmid was treated with 5, 10, 20 µM LM and the 20 μM camptothecin (CPT) was used as positive control. (**E**) The quantification of electrophoretic band. The Y axis is SC DNA/total DNA. (**F**) The images of tumors from each group. Tumor-bearing mice were administered the vehicle (negative control), 15 mg/kg Paclitaxel (positive control), LM (10 or 20 mg/kg per day). (**G**) The average tumor weight in each group. Data are presented as the mean ± S.D. *n* = 4, * *p* < 0.05 and ** *p* < 0.01, vs. negative control. (**H**) The average tumor volume in each group recorded during the treatments. Data are presented as the mean ± S.D., *n* = 4, * *p* < 0.05 and ** *p* < 0.01, vs. negative control. (**I**) The average body weight in each group. (**J**) The HE staining on stripped tumor tissue.

**Figure 2 molecules-27-06958-f002:**
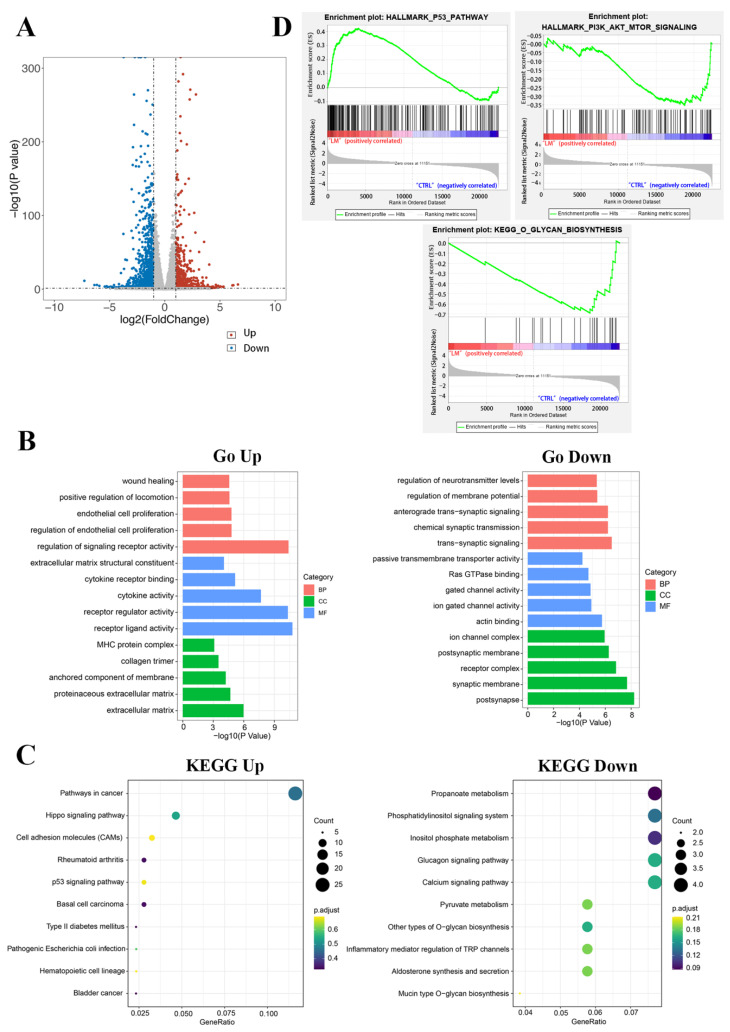
Functional analysis of differential genes. (**A**) Scatter plots of differentially expressed genes. There were 737 up-regulated genes and 1027 down-regulated genes in the LM group. Abscissa is the difference multiple (logarithmic transformation based on 2); (**B**) The gene ontology annotation analysis between DAGs and classification of BP, MF, and CC; (**C**) The KEGG pathway analysis of related DAGs; (**D**) The GSEA results of differentially abundant genes.

**Figure 3 molecules-27-06958-f003:**
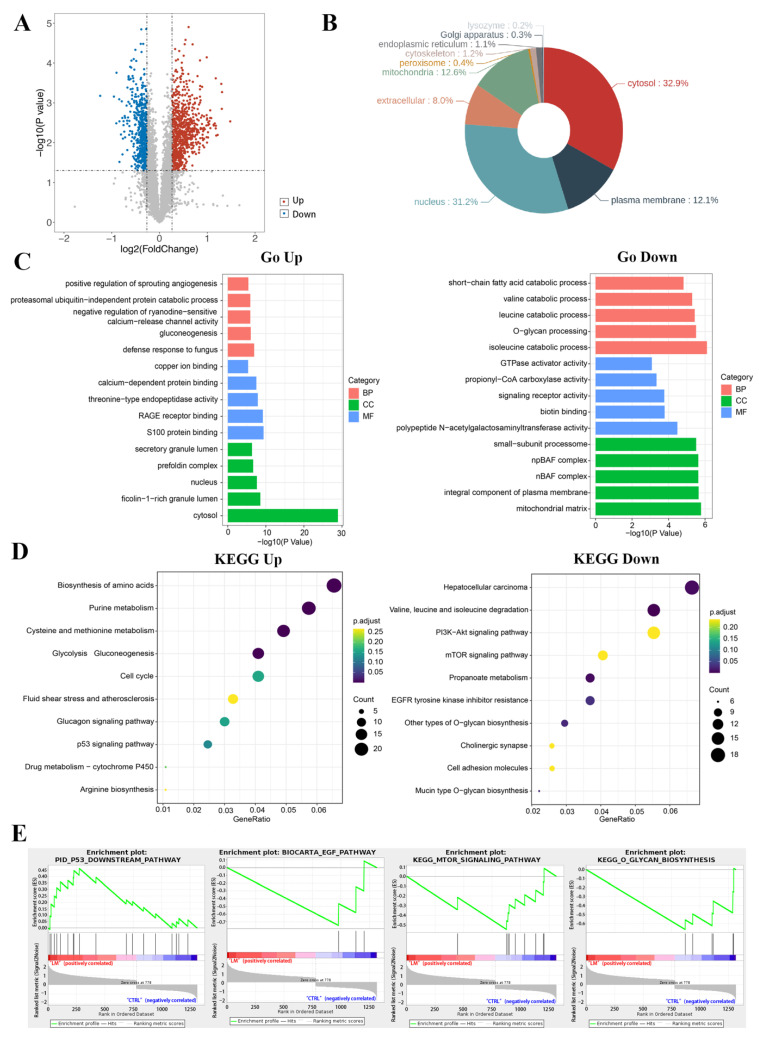
Functional analysis of differential proteins. (**A**) Scatter plots of differentially expressed proteins. There were 778 up-regulated proteins and 536 down-regulated proteins in the LM group. Abscissa is the difference multiple (logarithmic transformation based on 2); (**B**) localization analysis of differentially expressed proteins; (**C**) the gene ontology annotation analysis between DAPs and classification of BP, MF, and CC; (**D**) the KEGG pathway analysis of related DAPs; (**E**) the GSEA results of differentially abundant proteins.

**Figure 4 molecules-27-06958-f004:**
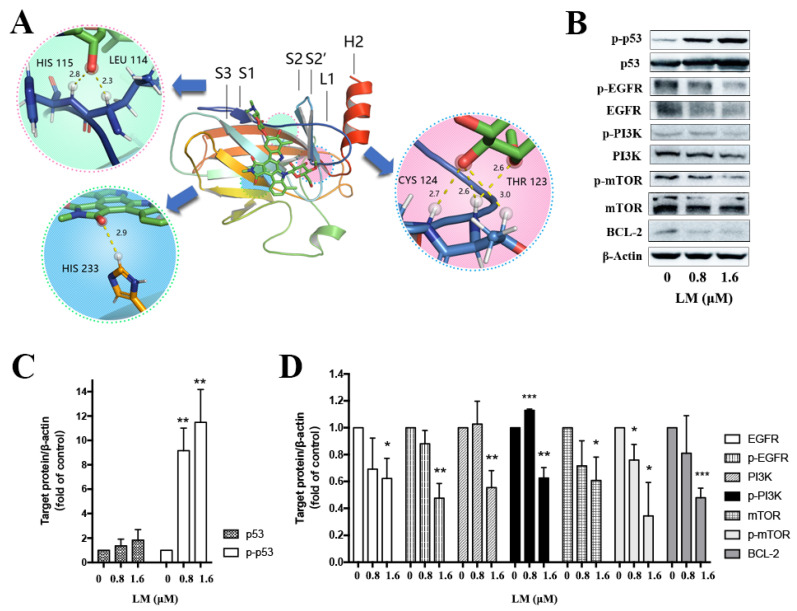
Validation of key proteins based on combined transcriptomic and proteomic results. (**A**) The docking picture of LM and p53 (L1/S3). (**B**) Western blot of key differentially abundant proteins. The MDA-MB-468 cells were treated with LM by 0, 0.8 and 1.6 μM. (**C**,**D**) Fold change in protein levels (LM treatment group/control group) from Western blot analysis. Significance: * *p* < 0.05, ** *p* < 0.01, *** *p* < 0.001 versus the control, *p* value based on *t*-test.

**Table 1 molecules-27-06958-t001:** IC_50_ value of LM on multiple tumor cell lines in 48 h.

Cells	IC_50_ (µM)	Cells	IC_50_ (µM)
L-02	1.022 ± 0.084	L-02	1.02 ± 0.084
MDA-MB-468	0.372 ± 0.201	PANC-1	0.591 ± 0.091
MDA-MB-231	0.197 ± 0.043	SMMC-7721	0.623 ± 0.044
A549	0.578 ± 0.096	HeLa	0.609 ± 0.037
SH-SY5Y	0.629 ± 0.028	NCI-H446	0.664 ± 0.062
PC-3	0.689 ± 0.102	U251	0.935 ± 0.205
MCF-7	0.517 ± 0.146	SW1990	1.30 ± 0.022

## Data Availability

The original contributions presented in the study are included in the article, further inquiries can be directed to the corresponding author.

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
