# Peer review of "Transcriptomics and Proteomics Characterizing the Anticancer Mechanisms of Natural Rebeccamycin Analog Loonamycin in Breast Cancer Cells"

_molecules, 2022, doi:10.3390/molecules27206958_

Round 1
Reviewer 1 Report
The study presented here aims to evaluate the effect of loonamycin (LM) on the MDA-MB-468 breast cancer cell line. The authors first evaluated the cytotoxic effect of LM and then used transcriptomics and proteomics to identify key players in LM's mechanism of action.
The study is of interest, but the incompleteness of the material and methods and poor quality of the figures (small text and pixelated) prevent an adequate evaluation of the methods and significance of the findings. The manuscript would also benefit from extensive editing of English language, although it is to note that the authors are unlikely native english speakers.
Major Comments:
1. Incomplete material and methods section prevents the adequate assessment of the work presented:
- No information provided on the statistical tests used (e.g. Fig 1).
- To my knowledge and according to the cellosaurus, MDA-MB-468 is a human cell line. Yet, according to the material and methods, RNA-seq reads were mapped to the Rat (Rattus norvegicus) genome. Please validate.
Additionnally, the material and methods should state which genome version was used in order to help reproducibility.
- No information is given on the LC-MS/MS run in the material and methods.
- The database used for proteomic analysis is not mentioned in the material and methods, neither does it include crucial parameters chosen for the analysis (such as PTMs, miscleavages, enzyme, etc...)
- No information is provided on the molecular docking method in the material and methods
2. The method for differential protein expression from TMT data is not described in the material and methods. The authors only mention the use of DESeq2 (which is a R package for differential expression analysis of the RNA-seq data).
This suggests that authors may have used DESeq2 for their proteomic data (?), yet this poses a lot of problems:
- how did the authors account for the fact that protein abundance data are continuous instead of counts for RNAseq data (which is crucial for the validity of negative binomial variance modeling).
- Publications using DESeq2 for proteomics data did so using spectral count from unlabelled data (counts), which is not the case here since the authors used TMT-labelling (continuous data).
- For labelled data, the authors should use Bayesian models (limma package in R) or other adequate statistics.
3. The figures are unreadable. It is to bear in mind that this may be due to a formatting problem at the time of submission, but it prevents independent evaluation of the data presented.
- The color legend on Figure 1C is unreadable (too small and pixelated)
- Quality of Figure 2 and 3 is poor and legends are unreadable.
4. The authors sometimes overstates their findings:
- The authors claim preferential LM cytotoxicity on TNBC cell lines. However, the dose-response curve for the control cell line is not shown which prevent the assessment of such claim by the reader.
- "these data suggest that LM exhibits better therapeutic activity": since LM has not been directly compared with other drugs in this study, this claim is unjustified. LM activity can be deemed promising, but any notion of comparison would have to be substantiated with experimental data.
5. The authors used the L02 cell line as control. L02 is a liver cell line, with different characteristics from breast cell lines. The authors would have a more compelling claim using a normal breast cell line such as MCF10A, especially since L02 is a known problematic cell line with cancerous characteristics (PMID 27349837) and proven contamination (PMID 26116706).
6. For the differential Gene expression analysis, the authors chose a leniant threshold of p < 0.05 which results in numerous hits (>1000). As we can see on the volcano plot (Fig 2A), this threshold is not very discriminant.
This creates noise in subsequent analyses (such as the GO enrichment analysis). The work would be more insightful by using a more stringent threshold p < 0.01 or p < 0.001.
Minor comments:
1. line 64 should refer to Fig 1D
2. Although the Figure 2 is almost unreadable, in Fig 2C one can read calcium signalign pathway in the KEGG-up graph and p53 signaling pathway in the KEGG-down graph, yet the text (lines 104-108) states the opposite?
Author Response
Answers to Reviewer #1’s Comments:
Comment 1: 1. Incomplete material and methods section prevents the adequate assessment of the work presented: No information provided on the statistical tests used (e.g. Fig 1).
Answer 1: This part of information was added in Materials and Methods 4.11.
“One-way ANOVA analysis was used for comparison between groups, and T-test was used for pair-wise comparison within groups. All statistical analyses were processed with GraphPad Prism 7.0 software (GraphPad Software, USA). P < 0.05 was considered statistically significant.”
Comment 2: To my knowledge and according to the cellosaurus, MDA-MB-468 is a human cell line. Yet, according to the material and methods, RNA-seq reads were mapped to the Rat (Rattus norvegicus) genome. Please validate. Additionnally, the material and methods should state which genome version was used in order to help reproducibility.
Answer 2: It was a mistake in writing. The genome version was Human genes (GrCH38.p12). This part of the information has been corrected in Materials and Methods 4.5.
Comment 3: No information is given on the LC-MS/MS run in the material and methods.
Answer 3: This part of information was added in Materials and Methods 4.6.
“Each fraction was injected for nanoLC-MS/MS analysis. The peptide mixture was loaded onto the C18-reversed phase analytical column (Thermo Fisher Scientific,Acclaim PepMap RSLC 50um X 15cm, nano viper,P/N164943) in buffer A (0.1% Formic acid) and separated with a linear gradient of buffer B (80% acetonitrile and 0.1% Formic acid) at a flow rate of 300 nl/min. The linear gradient was as follows:6% buffer B for 3 min, 6-28% buffer B for 42 min, 28-38% buffer B for 5min,38-100% buffer B for 5 min , hold in 100% buffer B for 5 min.
LC-MS/MS analysis was performed on a Q Exactive HF mass spectrometer (Thermo Fisher Scientific) that was coupled to Easy nLC (Thermo Fisher Scientific) for 60 min. The mass spectrometer was operated in positive ion mode. MS data was acquired using a data-dependent top10 method dynamically choosing the most abundant precursor ions from the survey scan (350–1800 m/z) for HCD fragmentation. Survey scans were acquired at a resolution of 60000 at m/z 200 with an AGC target of 3e6 and a maxIT of 50 ms. MS2 scans were acquired at a resolution of 15000 for HCD spectra at m/z 200 with an AGC target of 2e5 and a maxIT of 45 ms, and isolation width was 2 m/z. Only ions with a charge state between 2-6 and a minimum intensity of 2e3 were selected for fragmentation. Dynamic exclusion for selected ions was 30s. Normalized collision energy was 30 eV.”
Comment 4: The database used for proteomic analysis is not mentioned in the material and methods, neither does it include crucial parameters chosen for the analysis (such as PTMs, miscleavages, enzyme, etc...)
Answer 4: This part of information was added in Materials and Methods 4.6.
“MS/MS raw files were processed using MASCOT engine (Matrix Science, London, UK; version 2.6) embedded into Proteome Discoverer 2.2. The protein database was Uniprot_HomoSapiens_20367_20200226.The search parameters included trypsin as the enzyme used to generate peptides with a maximum of 2 missed cleavages permitted. A precursor mass tolerance of 10 ppm was specified and 0.05 Da tolerance for MS2 fragments. Except for TMT labels, carbamidomethyl (C) was set as a fixed modification. Variable modifications were Oxidation(M) and Acetyl (Protein N-term). A peptide and protein false discovery rate of 1% was enforced using a reverse database search strategy. Proteins with Fold change>1.2 and p value (Student’s t test) <0.05 were considered differentially expressed proteins. Wolf PSORT software was used for localization analysis of differential proteins. Gene Ontology analysis (GO) and KEGG pathway enrichment analysis of DEGs were implemented by clusterProfiler (3.4.4). GSEA enrichment analysis was performed using GSEA (V4.1.0)”
Comment 5: No information is provided on the molecular docking method in the material and methods
Answer 5: This part of information was added in Materials and Methods 4.10.
“Molecular docking was performed using the Schrodinger software (Schrödinger, Inc., New York, NY, USA). The 3D structure of p53 were retrieved from the protein data bank (PDB ID: 1TSR). In LM-p53 covalent docking, the protein p53 was prepared by Protein Preparation Tool in Schrodinger including optimized hydrogen bond network at pH 7.0 with PROKA tool. The ligand LM was prepared by Avogadro Tool to obtain the structural optimization.”
Comment 6: The method for differential protein expression from TMT data is not described in the material and methods. The authors only mention the use of DESeq2 (which is a R package for differential expression analysis of the RNA-seq data). This suggests that authors may have used DESeq2 for their proteomic data (?), yet this poses a lot of problems:
- how did the authors account for the fact that protein abundance data are continuous instead of counts for RNAseq data (which is crucial for the validity of negative binomial variance modeling).
- Publications using DESeq2 for proteomics data did so using spectral count from unlabelled data (counts), which is not the case here since the authors used TMT-labelling (continuous data).
- For labelled data, the authors should use Bayesian models (limma package in R) or other adequate statistics.
Answer 6: To these above questions, it was a mistake in writing. DESeq2 was used for RNA-seq data, not for TMT data. The MS/MS raw files were processed using MASCOT engine (Matrix Science, London, UK; version 2.6) embedded into Proteome Discoverer 2.2,
The method for differential protein expression from TMT data was added in Materials and Methods 4.6. And the other information has been corrected in Materials and Methods 4.6, too.
Comment 7: The figures are unreadable. It is to bear in mind that this may be due to a formatting problem at the time of submission, but it prevents independent evaluation of the data presented. - The color legend on Figure 1C is unreadable (too small and pixelated) - Quality of Figure 2 and 3 is poor and legends are unreadable.
Answer 7: The Figure 1, Figure 2 and Figure 3 have been recreated.
Comment 8: 4. The authors sometimes overstates their findings:
- The authors claim preferential LM cytotoxicity on TNBC cell lines. However, the dose-response curve for the control cell line is not shown which prevent the assessment of such claim by the reader.
Answer 8: In this part, we found that the IC50 of several breast cancer cell lines were below 0.6 mM, which may display good cytotoxicities as an anticancer lead compound. The reviewer gave us the good suggestion. We use “preferential” inappropriate, overstating the effect of LM. We corrected it as “good”.
Comment 9: "these data suggest that LM exhibits better therapeutic activity": since LM has not been directly compared with other drugs in this study, this claim is unjustified. LM activity can be deemed promising, but any notion of comparison would have to be substantiated with experimental data.
Answer 9: Thank you for the reviewer’s good suggestion. In our experiment, LM indeed exhibited good activity against TNBC, but it’s not directly compared with other positive drugs, which indicated LM is a promising anticancer lead compound. In our paper, the statement is not rigorous. We have changed it as follows: “Taken together, these data suggested that LM exhibits good therapeutic activity.”
Comment 10: The authors used the L02 cell line as control. L02 is a liver cell line, with different characteristics from breast cell lines. The authors would have a more compelling claim using a normal breast cell line such as MCF10A, especially since L02 is a known problematic cell line with cancerous characteristics (PMID 27349837) and proven contamination (PMID 26116706).
Answer 10: As for the application of L02 cell line as control, we initially aimed to roughly evaluate the effect of LM on normal cells. Although the application of L02 cells is controversial, we can still find researches about L02 as a normal cell line [1-3]. We will take your suggestions and select more suitable cell lines in the future research.
[1] Guo H, Ruan C, Zhan X, et al. Crocetin Protected Human Hepatocyte L02 Cell From TGF-β-Induced Oxygen Stress and Apoptosis but Promoted Proliferation and Autophagy via AMPK/m-TOR Pathway[J]. Front Public Health, 2022, 10: 909125.
[2] Wang Y, Xu L, Peng L, et al. Polybrominated diphenyl ethers quinone-induced intracellular protein oxidative damage triggers ubiquitin-proteasome and autophagy-lysosomal system activation in L02 cells[J]. Chemosphere, 2021, 275: 130034.
[3] Geng S, Wang S, Zhu W, et al. Curcumin suppresses JNK pathway to attenuate BPA-induced insulin resistance in L02 cells[J]. Biomed Pharmacother, 2018, 97: 1538-1543.
Comment 11: For the differential Gene expression analysis, the authors chose a leniant threshold of p < 0.05 which results in numerous hits (>1000). As we can see on the volcano plot (Fig 2A), this threshold is not very discriminant. This creates noise in subsequent analyses (such as the GO enrichment analysis). The work would be more insightful by using a more stringent threshold p < 0.01 or p < 0.001.
Answer 11: The used p value was the corrected p value. We think that p < 0.05 has certain reference significance. And the conclusion has been confirmed by WB methods.
Comment 12: Minor comments:
line 64 should refer to Fig 1D
Answer 12: It was a mistake in writing and was corrected in the text.
Comment 13: Although the Figure 2 is almost unreadable, in Fig 2C one can read calcium signaling pathway in the KEGG-up graph and p53 signaling pathway in the KEGG-down graph, yet the text (lines 104-108) states the opposite?
Answer 13: The diagram was misplaced and was corrected in Figure 2.
Reviewer 2 Report
Sun et. al., have tried to decipher the anticancer mechanism of the Loonamycin in breast cancer cells. The present work is interesting; however, reviewing the paper in the current format is highly challenging. Most figures are extremely low resolution, where nothing can be understood about what is being represented. The texts inside the figures are almost invisible. The authors are incredibly careless in referring to the figures (for example, in sections 2.2, 2.3, there is no reference for which data figure they are describing). In some cases the fig reference is wrong; for example, in page 2, line 71, Fig2A-2D reference is wrong; on page 5, line 162, figure ref 4D is wrong). Without these significant improvements, it's impossible to judge the merit of this paper.
Author Response
Answers to Reviewer #2’s Comments:
Comment 1: Sun et. al., have tried to decipher the anticancer mechanism of the Loonamycin in breast cancer cells. The present work is interesting; however, reviewing the paper in the current format is highly challenging. Most figures are extremely low resolution, where nothing can be understood about what is being represented. The texts inside the figures are almost invisible. The authors are incredibly careless in referring to the figures (for example, in sections 2.2, 2.3, there is no reference for which data figure they are describing). In some cases the fig reference is wrong; for example, in page 2, line 71, Fig2A-2D reference is wrong; on page 5, line 162, figure ref 4D is wrong). Without these significant improvements, it's impossible to judge the merit of this paper.
Answer 1: Thank you for the reviewer’s suggestion. The main comment is about the quality of the figures. We have reorganized the figures 1-4 to meet the quality of the journal. We hope that it will be helpful in reviewing the manuscript.
Round 2
Reviewer 2 Report
Sun et al. have tried to find out the functional mechanism of the anticancer drug Loonamycin (LM) in breast cancer cells. The present is intriguing; however, the authors need to address several points to improve the manuscript.
1. Although the authors have included images with better resolution than the initial submission, the authors still need to provide images with at least 300 dpi resolution so that each letter/feature of any figure can be easily visible in a printed copy. For example, the error bar numbers in Fig 1C, writings inside fig 2D, fig 3C, and 3D are still not clearly readable.
2. The authors have shown here that the LM influences the p53 pathways. The authors need to perform an extensive literature survey and discuss id this p53 pathway alteration is very common for different cancer drugs or it is selective for this LM compound. How different is LM’s functional mechanism compared to the rebeccamycin analogs, becatecarin, edotecacin, and NSC#655649?
3. “This compound showed an impressive cytotoxicity in vitro but could not be further developed because of poor water solubility.” Is it cytotoxic to any kind of cell or specific cancer cells? This needs to be clear in the main text.
4. “Three kinds of rebeccamycin analogues, becatecarin [5], edotecacin [6] and NSC#655649 [7] have entered clinical research.” How different are these analogs compared to LM?
5. “LM displayed a preferential anticancer activity against the cell lines with IC50 values of 0.517μM, 0.197 μM and 0.484 μM in MCF-7 cell line, MDA-56 MB-231 cell line and MDA-MB-468 cell line. The IC50 of LM for L02 was 1.022μM” The authors either need to show data or refer to a published paper.
6. In Fig 1B, negative controls are missing. The authors need to use a non-cancerous cell line to show that this effect is not generic. Also, the authors need to use a similar-structured different compound which can act as a negative control.
7. In Fig 1C, put the x-axis legend (i.e. mM) in the middle bottom portion of the axis.
8. The authors need to explain Fig 1D in the main text and also need to perform quantification.
9. The authors have used paclitaxel in Fig 1D. Is it a positive control? The authors need to cite a reference.
10. “The frozen section and HE staining on the stripped tumor tissue was performed, and obvious tumor-like tissues under the microscope was observed, such as large cell volume, big nucleus and deformity loose arrangement of tumor cells.” The authors need to make clear arrows to point in Fig 1G where these deformities are observed.
11. “There were no detectable toxic or necrotic effects on the heart, liver, spleen, lung or kidney tissues after LM treatment and no significant weight loss.” No data was shown.
12. “To uncover the LM regulatory mechanism in TNBC MDA-MB-468 cells, we performed RNA-seq analysis to profile the transcriptomes of MDA-MB-468 cells when treated with LM.” Mention the concentration of LM here.
13. “KEGG pathway enrichment analysis was conducted to describe the significant changes in pathways of DAGs.” Briefly describe in the main text how KEGG analysis is different from GO analysis.
14. “The results showed that the up-regulated differential genes were mainly centered on the pathways in cancer”, Are these gene cancer promoting or inhibiting?
15. The authors need to discuss in detail whether the proteomics analysis and gene analysis data are correlated or non-correlated.
16. The author’s data suggested that LM influences the p53 pathways and the authors have done docking studies with it. However, there is no direct evidence that LM interacts with p53. There could be many, many ways LM can influence the p53 pathways without directly interacting with it. What is the basis for assuming LM directly interacts with p53?
17. The authors have shown that LM strongly influences the upregulation of p-p53; however, there is no explicit discussion about how the phospho version is more upregulated.
18. On page 10, lines 238-258 seem redundant. This portion needs to have a proper linkage and should be shortened to one-third of the present form.
19. The authors need to do an extensive English language checking, for example,
“As far as we knew, rebeccamycin is a cytotoxicity compound which bound to the topoisomerase I to inhibit the reconnection at the DNA strand incision.”
It should be,
As far as we know, rebeccamycin is a cytotoxic compound that binds to topoisomerase I to inhibit the reconnection at the DNA strand incision.
“It’s demonstrated that EGFR protein expression is more overexpressed in TNBC, varying greatly from 13% to 76%” correct the English.
Author Response
The Confirmation List of the Revised Manuscript
Journal: Molecules
Title: Transcriptomics and Proteomics Characterizing the Anticancer Mechanisms of Natural Rebeccamycin Analog Loonamycin in Breast Cancer Cells (molecules-1868604)
The Reviewer’s comments are so beneficial and helpful to our manuscript. We have carefully checked this paper according to all their constructive suggestions and answered all the questions raised in the comments.
Answers to Reviewer #1’s Comments:
Comment 1: Although the authors have included images with better resolution than the initial submission, the authors still need to provide images with at least 300 dpi resolution so that each letter/feature of any figure can be easily visible in a printed copy. For example, the error bar numbers in Fig 1C, writings inside fig 2D, fig 3C, and 3D are still not clearly readable.
Answer 1: The modified resolution of Fig 1 is 300 dpi and Fig 2&3 are 500 dpi. The Fig. 1C, 2D, 3C, and 3D has been adjusted.
Comment 2: The authors have shown here that the LM influences the p53 pathways. The authors need to perform an extensive literature survey and discuss id this p53 pathway alteration is very common for different cancer drugs or it is selective for this LM compound. How different is LM’s functional mechanism compared to the rebeccamycin analogs, becatecarin, edotecacin, and NSC#655649?
Answer 2:The related elaboration of the first question is discussed in the second paragraph of the discussion. p53 signal pathway is a classical pathway of cancer occurrence, and is very common for different cancer treatment. The main pharmacological effect of rebeccamycin and its analogues is to inhibit the activity of topoisomerase, and cause the breakage of DNA molecules and block the cell cycle[1-3].
Rebeccamycin is an antitumor antibiotic and it inhibits DNA topoisomerase I. Rebeccamycin appears to exert its primary antineoplastic effect by poisoning topoisomerase I and has negligible effect on protein kinase C and topoisomerase II[4, 5].Becatecarin is a rebeccamycin analog with antitumor effects. Becatecarin intercalates into DNA and inhibites the catalytic activity of topoisomerases I/II[6]Edotecarin is a potent inhibitor of topoisomerase I that can induces single-strand DNA cleavage,and also has effect on protein kinase C[7]. From our experiment, we found that LM can effectively inhibit topoisomerase I.
Some studies have found that rebeccamycin can activate ATM, ATR, and activate downstream molecule Chk1 through ATR-Chk1 pathway. And then further block the cell cycle, so as to achieve the anti-tumor effect[8-13].In our experiment, we detected the protein expression of ATR-Chk1 pathway, and found that LM had no obvious effect on this pathway, so it was not mentioned in the article. What we can confirm is that LM can inhibit the activity of topoisomerase I and block cell cycle, but not by acting on ATR-Chk1 pathway.
Comment 3: “This compound showed an impressive cytotoxicity in vitro but could not be further developed because of poor water solubility.” Is it cytotoxic to any kind of cell or specific cancer cells? This needs to be clear in the main text.
Answer 3: L02 was used as a control cell line. It’s shown that LM was less toxic to L02 than any other cancer cell line, of which LM was sensitive to the MDA-MB-231 and MDA-MB-468 TNBC cell lines. The relevant statements is in the first paragraph "2.1 LM targets TNBC in vitro and in vivo " of the text.
Comment 4: “Three kinds of rebeccamycin analogues, becatecarin [5], edotecacin [6] and NSC#655649 [7] have entered clinical research.” How different are these analogs compared to LM?
Answer 4: The relevant explanations have been answered in comment 2.
Comment 5: “LM displayed a preferential anticancer activity against the cell lines with IC50 values of 0.517μM, 0.197 μM and 0.484 μM in MCF-7 cell line, MDA-MB-231 cell line and MDA-MB-468 cell line. The IC50 of LM for L02 was 1.022μM” The authors either need to show data or refer to a published paper.
Answer 5: These are the conclusions of the experiment, mentioned in "2.1 LM targets TNBC in vitro and in vivo".
Comment 6: In Fig 1B, negative controls are missing. The authors need to use a non-cancerous cell line to show that this effect is not generic. Also, the authors need to use a similar-structured different compound which can act as a negative control.
Answer 6: Fig. 1B shows that LM has a certain inhibitory effect on the proliferation of MDA-MB-468. In the experiment, we used L02 as negative control of normal cells, and compared the IC50 value, showing that LM was less toxic to L02. We just wanted to express that LM has certain effect on the proliferation of MDA-MB-468, no cytotoxicity experiment of similar structured compounds was conducted.
Comment 7: In Fig 1C, put the x-axis legend (i.e. mM) in the middle bottom portion of the axis.
Answer 7: The figure has been corrected in the text.
Comment 8: The authors need to explain Fig 1D in the main text and also need to perform quantification.
Answer 8: This part has been modified in the text.
In the electrophoresis of plasmids, the fastest is the super helical DNA, the second is the linear DNA, and the slowest is the open circular DNA. The topoisomerase I was added into the plasmids, which will destroy the structure of the super helix DNA. Camptothecin (CPT) is a classical topoisomerase I inhibitor, so it is used as a positive control. From the super helix DNA bands in the figure, LM can obviously inhibit the activity of topoisomerase I.
Comment 9: The authors have used paclitaxel in Fig 1D. Is it a positive control? The authors need to cite a reference.
Answer 9: Paclitaxel is a positive control[14]. This part has been modified and explained in the text.
Comment 10: “The frozen section and HE staining on the stripped tumor tissue was performed, and obvious tumor-like tissues under the microscope was observed, such as large cell volume, big nucleus and deformity loose arrangement of tumor cells.” The authors need to make clear arrows to point in Fig 1G where these deformities are observed.
Answer 10: We think there’s typical large cell volume and large nucleus with abnormal and deformed loose arrangement in the mass of the histiocytes in the figures. It's hard to point out specific one or two deformities.
Comment 11: “There were no detectable toxic or necrotic effects on the heart, liver, spleen, lung or kidney tissues after LM treatment and no significant weight loss.” No data was shown.
Answer 11: After the animal experiment, we dissected the experimental mice and found no damage on the heart, liver, spleen, lung and kidney tissues. The date of weight change has been posted in Fig.1F.
Comment 12: “To uncover the LM regulatory mechanism in TNBC MDA-MB-468 cells, we performed RNA-seq analysis to profile the transcriptomes of MDA-MB-468 cells when treated with LM.” Mention the concentration of LM here.
Answer 12: The concentration of LM has been modified in the text.
Comment 13: “KEGG pathway enrichment analysis was conducted to describe the significant changes in pathways of DAGs.” Briefly describe in the main text how KEGG analysis is different from GO analysis.
Answer 13: This part has been modified in “2.2 Functional annotation enrichment of LM-regulated genes”. GO (Gene ontology) is a comprehensive database that describes gene functions. It annotates gene products by 3 parts: cell component (CC), molecular function (MF), and biological process (BP). Through GO analysis, we can roughly understand which biological functions, signal pathways, or cell locations are enriched of the DEGs (DAGs). KEGG (Kyoto Encyclopedia of Genes and Genomes) is a comprehensive database integrating genome, chemistry and system function information. It stores information on gene pathways of different species. Through comparison, it can locate on the biological functions or signal pathways of the gene set may be concentrated.
Comment 14: “The results showed that the up-regulated differential genes were mainly centered on the pathways in cancer”, Are these gene cancer promoting or inhibiting?
Answer 14: “pathways in cancer” is a term in KEGG pathway enrichment analysis. As the genes were obtained by date analysis after the MDA-MB-468 cells treated with LM, these genes are theoretically cancer inhibited. However, except the terms that are verified in this text, no further research has been carried out on the other terms listed.
Comment 15: The authors need to discuss in detail whether the proteomics analysis and gene analysis data are correlated or non-correlated.
Answer 15: According to the analysis of RNA-seq, the p53 and downstream signal pathway were up-regulated, and O-glycan and PI3K-AKT-mTOR signaling pathway were down-regulated in LM treatment group. According to the TMT-based quantitative proteomic, p53 and downstream signal pathway were up-regulated, and the EGFR and mTOR related pathway, and O-glycan were down-regulated in LM treatment group. It’s suggested that LM may have a significant effect on O-glycan, p53-related signal pathway and EGFR/PI3K/AKT/mTOR signal pathway in enrichment of the KEGG pathway. The relevant explanations were in the text.
Comment 16: The author’s data suggested that LM influences the p53 pathways and the authors have done docking studies with it. However, there is no direct evidence that LM interacts with p53. There could be many, many ways LM can influence the p53 pathways without directly interacting with it. What is the basis for assuming LM directly interacts with p53?
Answer 16: Indeed, LM can influence the p53 by many pathways. But in the combined RNA-seq and proteomics, there was no other information indicating the possible pathways. We just supposed that LM may directly interacted with p53. Through molecular docking, it’s shown that LM may interact with p53. There is no biological experiment to directly verify the interaction between them.
Comment 17: The authors have shown that LM strongly influences the upregulation of p-p53; however, there is no explicit discussion about how the phospho version is more upregulated.
Answer 17: Relevant explanations were in the second paragraph of the discussion. The stability and transcriptional activity of p53 depend on its phosphorylation [15]. According to the research, the phosphorylation of p53 protein Ser15, Ser20, Ser33 and Ser37 could enhance its binding to P300, thereby activating the transcriptional activity of p53.The phosphorylation of Thr18 can not only enhance p53-p300 binding, but also interfere with p53-MDM2 binding[16]
Comment 18: On page 10, lines 238-258 seem redundant. This portion needs to have a proper linkage and should be shortened to one-third of the present form.
Answer 18: This part has been simplified in the text.
Comment 19:The authors need to do an extensive English language checking, for example,
“As far as we knew, rebeccamycin is a cytotoxicity compound which bound to the topoisomerase I to inhibit the reconnection at the DNA strand incision.”
It should be, As far as we know, rebeccamycin is a cytotoxic compound that binds to topoisomerase I to inhibit the reconnection at the DNA strand incision.
“It’s demonstrated that EGFR protein expression is more overexpressed in TNBC, varying greatly from 13% to 76%” correct the English.
Answer 19: We have done an English language checking and this part has been modified as suggested.
References:
[1] Arimondo P B, Moreau P, Boutorine A, et al. Recognition and cleavage of DNA by rebeccamycin- or benzopyridoquinoxaline conjugated of triple helix-forming oligonucleotides[J]. Bioorg Med Chem, 2000, 8(4): 777-784.
[2] Arimondo P B, Bailly C, Boutorine A S, et al. Triple helix-forming oligonucleotides conjugated to indolocarbazole poisons direct topoisomerase I-mediated DNA cleavage to a specific site[J]. Bioconjug Chem, 2001, 12(4): 501-509.
[3] Arimondo P B, Hélène C. Design of new anti-cancer agents based on topoisomerase poisons targeted to specific DNA sequences[J]. Curr Med Chem Anticancer Agents, 2001, 1(3): 219-235.
[4] Merchant J, Tutsch K, Dresen A, et al. Phase I clinical and pharmacokinetic study of NSC 655649, a rebeccamycin analogue, given in both single-dose and multiple-dose formats[J]. Clin Cancer Res, 2002, 8(7): 2193-2201.
[5] Bush J A, Long B H, Catino J J, et al. Production and biological activity of rebeccamycin, a novel antitumor agent[J]. J Antibiot (Tokyo), 1987, 40(5): 668-678.
[6] Robey R W, Obrzut T, Shukla S, et al. Becatecarin (rebeccamycin analog, NSC 655649) is a transport substrate and induces expression of the ATP-binding cassette transporter, ABCG2, in lung carcinoma cells[J]. Cancer Chemother Pharmacol, 2009, 64(3): 575-583.
[7] Saif M W, Diasio R B. Edotecarin: a novel topoisomerase I inhibitor[J]. Clin Colorectal Cancer, 2005, 5(1): 27-36.
[8] Watari A, Sakamoto Y, Hisaie K, et al. Rebeccamycin Attenuates TNF-α-Induced Intestinal Epithelial Barrier Dysfunction by Inhibiting Myosin Light Chain Kinase Production[J]. Cell Physiol Biochem, 2017, 41(5): 1924-1934.
[9] Messaoudi S, Anizon F, Peixoto P, et al. Synthesis and biological activities of 7-aza rebeccamycin analogues bearing the sugar moiety on the nitrogen of the pyridine ring[J]. Bioorg Med Chem, 2006, 14(22): 7551-7562.
[10] Watari A, Hasegawa M, Yagi K, et al. Checkpoint Kinase 1 Activation Enhances Intestinal Epithelial Barrier Function via Regulation of Claudin-5 Expression[J]. PLoS One, 2016, 11(1): e0145631.
[11] Goto H, Natsume T, Kanemaki M T, et al. Chk1-mediated Cdc25A degradation as a critical mechanism for normal cell cycle progression[J]. J Cell Sci, 2019, 132(2).
[12] Bahassi E M, Ovesen J L, Riesenberg A L, et al. The checkpoint kinases Chk1 and Chk2 regulate the functional associations between hBRCA2 and Rad51 in response to DNA damage[J]. Oncogene, 2008, 27(28): 3977-3985.
[13] Dai Y, Grant S. New insights into checkpoint kinase 1 in the DNA damage response signaling network[J]. Clin Cancer Res, 2010, 16(2): 376-383.
[14] Hu Y, Manasrah B K, Mcgregor S M, et al. Paclitaxel Induces Micronucleation and Activates Pro-Inflammatory cGAS-STING Signaling in Triple-Negative Breast Cancer[J]. Mol Cancer Ther, 2021, 20(12): 2553-2567.
[15] Yogosawa S, Yoshida K. Tumor suppressive role for kinases phosphorylating p53 in DNA damage-induced apoptosis[J]. Cancer Sci, 2018, 109(11): 3376-3382.
[16] Teufel D P, Bycroft M, Fersht A R. Regulation by phosphorylation of the relative affinities of the N-terminal transactivation domains of p53 for p300 domains and Mdm2[J]. Oncogene, 2009, 28(20): 2112-2118.
Round 3
Reviewer 2 Report
The authors have addressed most of the queries adequately. However, the authors still need to address a few minor points.
· For comment 5, the authors have that it is the conclusion from, “2.1 LM targets TNBC in vitro and in vivo”. But there is no data shown for all these cell lines. Please provide the data from where the IC50 values were calculated.
· For Figure 1, there are subfigures names with small caps a,b, etc. This may be confusing. Give separate names for all and explain them in the figure caption or use roman numerals.
· The authors need to clarify in the main text that there is no direct proof that LM is interacting with p53, but the authors have explored this possibility as a potential route to influence the p53 pathways.
Author Response
The Reviewer’s comments are so beneficial and helpful to our manuscript. We have carefully checked this paper according to all their constructive suggestions and answered all the questions raised in the comments.
Answers to Reviewer’s Comments:
Comment 1: For comment 5, the authors have that it is the conclusion from, “2.1 LM targets TNBC in vitro and in vivo”. But there is no data shown for all these cell lines. Please provide the data from where the IC50 values were calculated.
Answer 1: The data have been made in Table.1 and inserted in “2.1 LM targets TNBC in vitro and in vivo”.
Comment 2: For Figure 1, there are subfigures names with small caps a,b, etc. This may be confusing. Give separate names for all and explain them in the figure caption or use roman numerals.
Answer 2: The subfigures were named separately and explained in the figure caption.
Comment 3: The authors need to clarify in the main text that there is no direct proof that LM is interacting with p53, but the authors have explored this possibility as a potential route to influence the p53 pathways.
Answer 3: This part has been explained in the third paragraph of the discussion (line 270-271).